# Distribution and Activity of Sulfur-Metabolizing Bacteria along the Temperature Gradient in Phototrophic Mats of the Chilean Hot Spring Porcelana

**DOI:** 10.3390/microorganisms11071803

**Published:** 2023-07-14

**Authors:** Ricardo Konrad, Pablo Vergara-Barros, Jaime Alcorta, María E. Alcamán-Arias, Gloria Levicán, Christina Ridley, Beatriz Díez

**Affiliations:** 1Department of Molecular Genetics and Microbiology, Biological Sciences Faculty, Pontifical Catholic University of Chile, Santiago 8331150, Chile; r.a.konrad@gmail.com (R.K.); pvergar1@uc.cl (P.V.-B.); jaalcort@uc.cl (J.A.); christina.m.ridley@gmail.com (C.R.); 2Millennium Institute Center for Genome Regulation (CGR), Santiago 8370186, Chile; 3Department of Oceanography, University of Concepcion, Concepcion 4030000, Chile; mealcaman@uc.cl; 4Center for Climate and Resilience Research (CR)2, Santiago 8370449, Chile; 5Escuela de Medicina, Universidad Espíritu Santo, Guayaquil 0901952, Ecuador; 6Biology Department, Chemistry and Biology Faculty, University of Santiago of Chile, Santiago 9170022, Chile; gloria.levican@usach.cl

**Keywords:** sulfur oxidation/reduction, *sox*B, *apr*A, phototrophic hot spring mat, metagenomics

## Abstract

In terrestrial hot springs, some members of the microbial mat community utilize sulfur chemical species for reduction and oxidization metabolism. In this study, the diversity and activity of sulfur-metabolizing bacteria were evaluated along a temperature gradient (48–69 °C) in non-acidic phototrophic mats of the Porcelana hot spring (Northern Patagonia, Chile) using complementary meta-omic methodologies and specific amplification of the *apr*A (APS reductase) and *sox*B (thiosulfohydrolase) genes. Overall, the key players in sulfur metabolism varied mostly in abundance along the temperature gradient, which is relevant for evaluating the possible implications of microorganisms associated with sulfur cycling under the current global climate change scenario. Our results strongly suggest that sulfate reduction occurs throughout the whole temperature gradient, being supported by different taxa depending on temperature. Assimilative sulfate reduction is the most relevant pathway in terms of taxonomic abundance and activity, whereas the sulfur-oxidizing system (Sox) is likely to be more diverse at low rather than at high temperatures. Members of the phylum *Chloroflexota* showed higher sulfur cycle-related transcriptional activity at 66 °C, with a potential contribution to sulfate reduction and oxidation to thiosulfate. In contrast, at the lowest temperature (48 °C), *Burkholderiales* and *Acetobacterales* (both *Pseudomonadota*, also known as *Proteobacteria*) showed a higher contribution to dissimilative sulfate reduction/oxidation as well as to thiosulfate metabolism. *Cyanobacteriota* and *Planctomycetota* were especially active in assimilatory sulfate reduction. Analysis of the *apr*A and *sox*B genes pointed to members of the order *Burkholderiales* (*Gammaproteobacteria*) as the most dominant and active along the temperature gradient for these genes. Changes in the diversity and activity of different sulfur-metabolizing bacteria in photoautotrophic microbial mats along a temperature gradient revealed their important role in hot spring environments, especially the main primary producers (*Chloroflexota*/*Cyanobacteriota*) and diazotrophs (*Cyanobacteriota*), showing that carbon, nitrogen, and sulfur cycles are highly linked in these extreme systems.

## 1. Introduction

Sulfur is one of the six most abundant chemical elements on planet Earth [1]. Its terrestrial biogeochemical cycle is influenced by volcanic activity, decomposition of organic matter, and fossil fuel combustion [2]. In terrestrial hot springs, solubilization and surface release of metallic sulfides occur due to increased hydrostatic pressure when groundwater is heated in the proximity of magma [2,3]. Subsurface geochemistry highly influences the concentration of sulfur compounds, which can be found at high concentrations in acidic sulfur springs (pH < 3), but at low sulfate concentrations (<100 mg/L) in chloride hot springs with more neutral pH [4]. These sulfur species can be used as electron acceptors or donors by a variety of sulfur-metabolizing microorganisms [5].

Microbial communities of terrestrial hot springs can develop tight mats stratified longitudinally by physicochemical gradients generated in part by their metabolism [6,7]. Temperature and sulfur concentrations are major drivers of the structure of these microbial communities [8,9,10]. In the upper layers of non-acidic hot spring mats, oxygenic phototrophic *Cyanobacteriota* (former *Cyanobacteria*) and anoxygenic phototrophic *Chloroflexota* members dominate, assimilating and utilizing sulfur as part of their metabolism [7,11,12,13,14,15]. In the deeper layers of the mat, anaerobic microorganisms capable of dissimilative sulfate reduction metabolism thrive [16,17].

Several metabolic pathways are involved in the sulfur cycle [16], with assimilative sulfur reduction (ASR), dissimilative sulfur reduction (DSR), and sulfur oxidation (SOX) being the most studied. ASR is mostly related to the assimilation of sulfur as a substrate to synthesize biological molecules [16], whereas DSR and SOX are mainly associated with energy processes that use sulfur compounds during electron transfer-associated processes [16]. Since the rise of the omics era, next-generation sequencing has been used to study biological diversity related to the sulfur cycle [18,19], helping to understand new aspects of this cycle in the environment.

In hot springs, the diversity of sulfur-reducing bacteria (SRB) and sulfur-oxidizing bacteria (SOB) has previously been studied based on the 16S rRNA gene [20,21,22], as well as the functional marker genes *apr*A, *sox*B, and *dsr*A [9,23,24,25]. The *apr*A gene encodes an adenosine-5′-phosphosulfate (APS) reductase that is highly conserved among SRB and is also present in several SOBs that utilize the so-called reverse sulfate reduction pathway [26]. During the sulfate dissimilative reduction process, phosphorylated APS, which is generated by the heterodimeric enzyme ATP sulfurylase, is directly reduced to sulfite by APS reductase. Thus, the *apr*A gene has been useful for simultaneously determining the diversity of both SRB and SOB microbial groups [27,28]. Indeed, this has allowed the detection of a variety of SRB, such as the genus *Thermodesulfobacterium* [24]; photolitotrophic sulfur oxidizers and strict anaerobes of the phylum *Chlorobiota*; and chemolithotrophs of the classes Alpha-, Beta-, and Gamma-proteobacteria found in both terrestrial and submarine hot springs [29,30].

Conversely, the *sox*B gene, encoding *sox*B thiosulfohydrolase, is the most conserved component of the sulfur-oxidizing Sox system, and catalyzes dissimilative oxidation of thiosulfate and other sulfur compounds [31]. The Sox system has been described as an alternative oxidation pathway to the reverse sulfate reduction pathway and has been reported in several SOB, including *Chlorobiota*, *Pseudomonadota* (Alpha- and Gamma-proteobacteria), and *Acidithiobacillia* [32]. In environments such as hydrothermal vents and terrestrial hot springs, the *sox*B gene has been used as a complimentary marker to the *apr*A gene to determine SOB diversity [28,30,32,33,34].

However, to date, only a few studies have described the microbial community variability related to sulfur cycling along the temperature gradient of terrestrial hot springs [9,22,35]. In addition, the diversity and transcriptional activity of microbial communities have been overlooked in non-acidic thermal systems. Thus, the objective of this work was to address the knowledge gap related to the variation and activity of sulfur cycle-related organisms along a temperature gradient in non-acidic hot springs. Accordingly, we used a complementary combination of methodologies to analyze the genomic taxonomy, diversity, distribution, and transcriptional activity of the sulfur-metabolizing bacterial community in microbial mats from the Porcelana hot spring (northern Chilean Patagonia). The main results reveal the diverse functions of sulfur-cycling microorganisms in these thermophilic microbial communities, many of which have been previously described for their particular roles in the carbon and nitrogen cycles [36,37].

## 2. Materials and Methods

### 2.1. Sampling Site and Sample Collection

Microbial mat samples were obtained from Porcelana hot spring, Chile (42°27′29.1″ S–72°27′39.3″ W) at different points along the temperature gradient: 56.8, 61.2, 63.9, and 69.7 °C in March 2011; and 48, 58, and 66 °C in March 2013 [37,38]. Temperature, conductivity, dissolved oxygen, and pH were monitored in situ at all sampling sites using a multiparameter instrument (Oakton, Des Plaines, IL, USA; model 35607-85) [37,38]. Water samples (15 mL) were collected in March 2013 to determine the inorganic sulfur species. To inhibit the microbial and inorganic oxidation of hydrogen sulfide, water samples were stored with a 2 M zinc acetate solution (Winkler ™, Santiago de Chile, Chile), and the pH was adjusted to ≥9 using a 6 M NaOH solution.

### 2.2. Determination of Inorganic Sulfur Species in Porcelana Thermal Water

For the quantification of inorganic sulfur compounds, the protocol described by the American Public Health Association was followed [39]. The sulfide calibration curve (five points from 0.05 to 0.5 mg/L S^−2^) was prepared from a serial dilution of a standard 10 mg/mL Na_2_S solution (Sigma-Aldrich™, St. Louis, MO, USA). Sulfide was measured spectrophotometrically at 664 nm using the methylene blue method [40]. The sulfate calibration curve (five points from 5 to 40 mg/L) was prepared using a 1 g/L stock solution of anhydrous Na_2_SO_4_. Sulfate was measured by turbidimetry analysis using spectrophotometry at 420 nm [41]. The thiosulfate calibration curve (five points from 0.5 to 4 mg/L) was prepared using a standard 0.01 M sodium thiosulfate pentahydrate solution. Thiosulfate levels were determined by measuring absorbance at 350 nm [42].

### 2.3. DNA Extraction, PCR Amplification, Cloning, and Sequencing of the aprA and soxB Genes

Genomic DNA from the microbial mat samples was extracted for all temperatures and years (56.8, 61.2, 63.9, and 69.7 °C, March 2011; and 48, 58, and 66 °C, March 2013) as previously described [37,38]. PCR amplification of the *apr*A and *sox*B genes was performed using the primer sets *apr*A-1-FW/*apr*A-5-RV [43] and *sox*B-693-F/*sox*B-1446-R [34], respectively. Primer sequences and final PCR conditions using GoTaq^®^ DNA Polymerase (Promega, Madison, WI, USA) are provided in Appendix A. PCR products were purified using the Wizard^®^ Clean-up System kit (Promega™) and cloned with the pJET^®^1.2/blunt cloning kit (Thermo Scientific, Waltham, MA, USA). Plasmids were transformed into chemo-competent *Escherichia coli* DH5α following the manufacturer’s protocol. *Escherichia coli* DH5α colonies were verified by PCR using the same primers as noted above. Verified clones were subjected to a second colony-PCR using the pJetF/pJetR primers (Thermo Scientific). From 100 clones per sample, 20% of the PCR products of the expected size were sent for sequencing (Macrogen Inc., Seoul, South Korea). Identification and assignment for a given operational taxonomic unit (OTU) were performed for the *apr*A and *sox*B clones by comparing restriction fragment length polymorphism patterns using NlaIV and SduI restriction enzymes. These enzymes were selected based on a cutting simulation using A Plasmid Editor (v2.0.45) software performed on randomly selected *apr*A and *sox*B gene clone sequences, which resulted in specific patterns for both genes. Enzymatic digestion of the PCR products of the clones from each clone library was performed according to the manufacturer’s instructions (Thermo Scientific). Eighty percent of the *apr*A and *sox*B sequences were identified by restriction patterns.

### 2.4. Sulfur Metabolism Reconstruction from Publicly Available Porcelana Hot Spring Metagenomes and Metatranscriptomes

Three shotgun metagenomes and three corresponding metatranscriptomes obtained from microbial mat samples collected along the temperature gradient (48, 58, and 66 °C; March 2013) of the Porcelana hot spring were downloaded from the NCBI SRA (SRA ID: SRP104009) [36]. Trimming, assembly, binning, and refinement were performed according to the literature [44]. Briefly, reads were trimmed using Cutadapt [45] and assembled with SPAdes (v3.10.1) [46]. Metagenome-assembled genomes (MAGs) were then generated by binning with the metaWRAP binning module [47] using the following three binning software: metaBAT 2 (v2.12.1) [48], MaxBin2 [49], and CONCOCT (v1.1.0) [50]. MAGs with more than 50% completeness and less than 10% contamination were refined with the bin_refinement module of metaWRAP [47] and the refineM tool [51]. Genome quality was assessed with the CheckM (v1.0.18) tool [52] and genomic taxonomy was determined with the GTDB-tk (v0.3.2) software with databases R95 and R202 [53]. Several phyla names (e.g., *Pseudomonadota* and *Cyanobacteriota*) were updated to the last GTDB R214 version across the manuscript.

The protein sequences of the three assembled Porcelana metagenomes [44] were predicted using the software Prodigal [54]. Taxonomy was assigned based on that provided by GTDB-tk (v0.3.2) R95 software [53] for proteins belonging to the MAGs. In addition, the taxonomic identity of unbinned proteins was determined by the closest match against predicted proteins from the GTDB R95 database [51] using DIAMOND software (blastp, e-value = 1 × 10^−6^) [55]. Functional annotation was performed using eggNOG-mapper software [56,57].

Trimmed reads obtained from the three metagenomes and metatranscriptomes (48, 58, and 66 °C) from samples obtained in 2013 [36] were aligned to the gene set from 67 MAGs or from unbinned contigs coming from the metagenomes using bowtie2 (v2.3.4.1) (--end-to-end--sensitive-k) [58]. The per-gene count table was processed in R using the Tidyverse package (https://www.tidyverse.org/, accessed in March 2021). The analysis focused on three main pathways of the sulfur cycle: (i) assimilative sulfate reduction (ASR), which is involved in sulfur assimilation and includes genes of the *cys* family; (ii) dissimilative sulfate reduction, which uses sulfur species as electron acceptors and includes the *dsr* and *apr* genes; and finally, (iii) the Sox system, which comprises the Sox family of proteins involved in thiosulfate metabolism. To prevent overcounting of KEGG orthologs (Kos), the number of reads associated with each gene was divided by the number of Kos associated with the respective gene, then equality was distributed among the respective Kos [59], and finally normalized by quantiles. Kos related to the sulfur cycle were obtained from the KEGG website (pathway ID: KO00920). Normalized sulfur counts per function were summarized by KO per taxa in each sample, while genes were grouped and summed according to the sulfur-cycle step in which they were involved. Heatmaps were generated using the ggplot package in R.

### 2.5. Taxonomic Description and Phylogenetic Reconstruction

Sequences corresponding to the *apr*A (TIGRFAM accession: TIGR02061) and *sox*B (TIGRFAM accession: TIGR04486) genes were extracted using the hmmsearch tool of the HMMER (v3.2.1) software (http://hmmer.org/, accessed in November 2018) from the gene set predicted from the MAGs and unbinned contigs assembled from the three Porcelana metagenomes. The obtained sequences were checked against the nr and refseq_protein databases to confirm annotations, avoiding the recruitment of other protein sequences with shared domains. Taxonomic assignment of the *sox*B and *apr*A genes from the cloned OTUs produced in this study and the three metagenomic samples was performed by the closest BLASTP match against the GTDB R95 custom protein database [51]. The *apr*A and *sox*B gene sequences from clones and metagenomes, as well as their closest references, were then aligned using MUSCLE (v6.0.98) software [60]. Maximum-likelihood trees were generated with the Iqtree (v.1.5.5) software, using the TESTNEW option to choose the best substitution model and the non-parametric ultrafast bootstrap option with 10,000 replicates [61]. Phylogenetic reconstructions, node collapse, and tree rooting were managed with the iTOL web server [62].

## 3. Results and Discussion

### 3.1. Changes in Sulfur Compounds along the Temperature Gradient at Porcelana Hot Spring

The sulfide, sulfate, and thiosulfate levels of samples collected in 2013 at the Porcelana hot spring (pH = 6 to 7; temperature = 48 to 69.7 °C) were measured spectrophotometrically (See Material and Methods). Appendix A summarizes these results and also includes data retrieved from previous studies [37,38]. The low sulfate levels (i.e., <100 mg/L according to [4]) observed in Porcelana (56.02 to 66.24 mg/L) at 48, 58, and 66 °C were similar to those previously reported in other non-acidic or neutral chloride hot springs from the Tibetan Plateau [9], Japan [22,63], Iceland, and Yellowstone National Park, USA [17,64,65]. Sulfate levels decreased with distance from the source, specifically from 66.24 mg/L at 66 °C to 57.82 mg/L at 58 °C and 56.02 mg/L at 48 °C. These levels were similar to those previously reported at Porcelana (between 40.6 and 66.9 mg/L, [37]), suggesting a constant presence of sulfate over time. Sulfide levels remained between 1.13 and 1.51 mg/L, while thiosulfate increased from 4.42 to 8.09 mg/L with decreasing temperature (Appendix A), resembling values reported for other hot springs [64,66]. Changes in sulfur species concentrations with temperature, such as lower sulfate levels at lower temperatures, suggest dynamic biological activity related to the sulfur cycle [37,38].

### 3.2. Key Active Sulfur-Metabolizing Organisms and Pathways in the Sulfur Cycle of Porcelana Revealed by Meta-Omics

Since organisms potentially related to the sulfur cycle were found in Porcelana microbial mats [37,38], we analyzed the already available meta-omic data from the temperature gradient at 48, 58, and 66 °C [36,44] to obtain a general but detailed overview of the presence and activity of microorganisms involved in the sulfur cycle.

The sulfur-cycle-related gene abundances were quantified by read alignment to genes predicted from MAGs and unbinned contigs and then normalized and processed as described in Materials and Methods. Individual gene counts were grouped according to their order and phylum and associated pathway (pathways and KO IDs shown in Appendix A), revealing the presence of the pathway (Figure 1) and its transcriptional levels (Figure 2). ASR and DSR reactions are mentioned in the reductive way for formality, as some of these enzymatic reactions can function also in an oxidative way [67]. The main results show that the ASR pathway was the most abundant and active sulfite-to-sulfide reduction pathway at 48, 58, and 66 °C (Figure 1A and Figure 2A). This pathway was mainly associated with dominant primary producers in Porcelana, such as members of the order *Cyanobacteria*les, belonging to the phylum *Cyanobacteriota*, and the order Chloroflexales, belonging to the phylum *Chloroflexota*. In contrast, the DSR pathway was only detected as an active sulfite-to-sulfide reduction reaction pathway in the order *Burkholderiales* from the phylum *Pseudomonadota* (*Proteobacteria*, Figure 2B), suggesting lower (but still present) activity compared to ASR.

For sulfate-to-sulfite reactions, both ASR and DSR were present and active in Porcelana (Figure 1C,D and Figure 2C,D). In terms of ASR abundance and activity, sulfate-to-sulfite reactions were more variable than sulfite-to-sulfide reactions (Figure 1C and Figure 2C). *Cyanobacteriota* and *Chloroflexota*, the most important primary producers in the Porcelana hot spring [36], remain the most abundant and active taxa for both sulfate-to-sulfite and sulfite-to-sulfide reactions. *Cyanobacteriota* were mostly present and active at 48 and 58 °C, while *Chloroflexota* were present and active at 58 and 66 °C. In addition, some phyla, such as Myxococcota, *Pseudomonadota* (*Proteobacteria*), *Planctomycetota*, and Bacteroidota, were present in both sulfate-to-sulfite and sulfite-to-sulfide reactions along the temperature gradient but were slightly more abundant at 48 °C than at the higher temperatures (Figure 1 and Figure 2), suggesting their contribution in both pathways of the sulfur cycle. However, it is likely that sulfate-to-sulfite reactions are more diverse than sulfite-to-sulfide reactions, as several taxa were absent for the sulfite-to-sulfide reactions, such as the phylum Armatimonadota at lower temperatures, or Thermotogota at higher temperatures. In the case of DSR, most reads were associated with the *sat* gene (Appendix A), which functions in both ASR and DSR, and is required for the synthesis of APS [16,67], an intermediate of the sulfate-to-sulfite reaction. Therefore, these results alone cannot truly discriminate the major taxa in DSR. However, most of the reads belonged to *Cyanobacteriota*, *Chloroflexota*, and *Pseudomonadota* (*Proteobacteria*), suggesting that these phyla might be involved in the DSR pathway. It has been previously reported that some bacteria belonging to *Pseudomonadota*, which is one of the main phyla found in the Porcelana hot spring, have the classical sulfur-reducing DsrAB and a sulfur-oxidizing DsrAB variant named rDsrAB [67,68], which may suggest that sulfur-redox reactions are occurring in both directions.

The Sox system proved to be the least abundant and active pathway of those studied in Porcelana (Figure 1E and Figure 2E). The most abundant taxa showing Sox (Figure 1E) were *Chloroflexota*, *Pseudomonadota* (*Proteobacteria*), and Deinococcota. However, Sox activity changed along the temperature gradient, specifically, *Pseudomonadota* were the most active community members at 48 and 58 °C, while *Chloroflexota* were the most active community members at 66 °C. Additionally, the activity of Deinococcota was highest at 66 °C (Figure 2E). These results suggest that thiosulfate metabolic activity in the Porcelana phototrophic mats is carried out by several members of the bacterial community along the temperature gradient, being more taxonomically diverse at the phylum level at lower temperatures. This may be exemplified by the dominance of a *Gammaproteobacteria sox*B OTU detected at higher temperatures (see below). The lower sulfate levels far from the spring source could be explained by higher SOX activity at lower temperatures, resulting in increased thiosulfate downstream. Although we cannot exclude an effect of net ASR and DSR activity pathways on sulfate concentrations, this is not supported by the low variations in sulfide levels.

Interestingly, some “microbial dark matter” taxa in the hot spring (e.g., Binatota and Verrucomicrobiota) showed high activity at 66 °C, despite their low abundances at the DNA level (Figure 1 and Figure 2). This suggests the potential and still unknown importance of these phyla in the sulfur cycle. In particular, the presence of 16S rRNA gene sequences from the recently described phylum Verrucomicrobiota was reported in a sulfur-active environment [69], while some MAGs belonging to Binatota (also referred to as Candidatus UBP10 [51]) were reported encoding for several genes involved in the metabolism of methylated sulfur compounds [70], suggesting, along with our results, that these phyla could have active sulfur metabolisms. However, further research is needed to confirm the relevance of these less abundant microorganisms in the sulfur cycle.

### 3.3. Complementary Diversity Analysis of Sulfur-Metabolizing Organisms Using aprA and soxB Marker Genes

Available metagenomic data allowed us to obtain an overview of the key microbial players in the sulfur cycle of the Porcelana hot spring. However, the lack of biological replicates (i.e., our analysis was restricted to a single metagenome per sampled temperature) limited us to a higher level of resolution and confidence for applying alpha diversity analyses. For this reason, we applied an additional complementary methodology that employed a combination of two marker genes, such as those targeting the Sox system (*sox*B gene) and APS reductase (*apr*A gene). Differential recovery and classification of taxa at different taxonomic levels have been previously reported using these marker genes [30,33].

For the clone libraries of the seven Porcelana samples along the temperature gradient (48–69 °C), the alpha rarefaction curve plateaued for the *apr*A gene, but not for the *sox*B gene (Appendix A). For *sox*B, two OTUs recovered in all samples were annotated as a hypothetical protein and a malate synthase, respectively, and were not considered in the subsequent analyses. An average of 100 clones per sample were analyzed for both genes using the restriction fragment length polymorphism method (with 20% clone sequencing, see Materials and Methods; Appendix A). A total of six different OTUs were obtained for the *apr*A gene (Figure 3), where OTU-1 and OTU-5 were ubiquitous, and OTU-1 was the most abundant across the temperature gradient. Alpha diversity (according to the Simpson and Shannon indices) for this gene (Table 1) was higher and had higher evenness indices at 58 and 66 °C compared to all other temperatures. For the *sox*B gene, only two ubiquitous OTUs were obtained, in which OTU-1 was dominant at all temperatures except 48 °C (Table 1). For this gene, the Simpson index ranged from 0.3 to 0.5 at all temperatures, except at 61.2 °C, which showed the lowest diversity (0.147). Although similar diversities, as shown by the Shannon and Simpson indices, were obtained for the microbial communities of both markers (*apr*A and *sox*B) at all temperatures, a decrease in diversity was observed at temperatures close to 60 °C. However, more data are required to corroborate a correlation between diversity and temperature at the Porcelana hot spring.

Similar to Porcelana, diversity estimates for sulfur-metabolizing microorganisms in other hot springs were lower than those for the entire microbial community, as measured by 16S rRNA gene analysis (e.g., [71]). In Porcelana, the Simpson and Shannon indices for the *apr*A and *sox*B OTUs were similar to other hot springs, such as those in Tibet [9]. However, sulfur-metabolizing bacteria are more diverse in other environments, such as in freshwater ecosystems, which have higher *apr*AB gene richness [72], or in sediment samples from the China Sea, which show higher Shannon index values for the *dsr*B and *sox*B genes [73]. This suggests an important role for a few taxa in the generally less diverse and extreme thermal ecosystem.

### 3.4. Phylogenetic Affiliation and Abundance of Sulfur-Metabolizing OTUs and MAGs

The phylogenetic affiliation (Figure 3 and Figure 4) and relative abundances of the *apr*A and *sox*B sequences (Table 2) were estimated for each temperature and year for clone libraries. The two recovered OTUs of the *sox*B gene were assigned to the phylum *Pseudomonadota* (*Proteobacteria*), whereas OTUs of the *apr*A gene were distributed in four phyla: *Pseudomonadota*, GTDB Bacillota_B (former Firmicutes), *Nitrospirota* (Nitrospirae), and *Desulfobacterota* (Thermodesulfobacteria). Furthermore, proteins predicted from the MAGs and unbinned contigs of the Porcelana metagenomes at 48, 58, and 66 °C [36,44] were analyzed, providing additional data on the organisms involved in the sulfur cycle.

For the *apr*A sequences, two clades were observed, one was composed exclusively of SOB members (OTU-1 and OTU-3), and the other comprised SOB (OTU-2) and SRB members (OTU-4, OTU-5, and OTU-6) (Figure 3). OTU-1 dominated at all temperatures except 66 °C (Table 2). This OTU was related to sequences from the order *Burkholderiales* (class *Gammaproteobacteria*; Figure 3), particularly showing 98% identity to *Lautropia* sp. SCN 69-89 (Table 2). This genus has been reported to be abundant in other hot springs [29,74,75,76] and its genome was classified as a new genus (g_SCN-69-89) by the GTDB R95 [51]. Additionally, the *apr*A OTU-1 (Figure 3) was associated with metagenomic *apr*A sequences within two Porcelana MAGs potentially classified as a new genus within the family Burkholderiaceae (Appendix A), thereby reinforcing the polyphyletic character suggested for the genus *Lautropia*.

Conversely, OTU-2, also belonging to the order *Burkholderiales* (Figure 3), was the second most abundant sequence for the *apr*A gene, but it only dominated at 66 °C. For this OTU, the closest sequence representing a cultured organism was the isolate *Thioalkalivibrio* sp. HK1 (90% identity), whereas the closest sequence representing an uncultured organism corresponds to a MAG recovered from a groundwater metagenome (95% identity; GCA_004297625.1; [76]), which was suggested to be involved in the sulfur cycle [77,78] and form a new genus and family in the order *Burkholderiales*. Both OTU-1 and OTU-2 reveal potential SOB in new Burkholderiaceae genera that have not yet been cultivated. Furthermore, the OTU-2 clade suggests polyphyly among *apr*A sequences within *Gammaproteobacteria*, being more closely related to the GTDB phylum *Nitrospirota* than to the OTU-1 clade (Figure 3), which, according to other studies, could be explained by extensive horizontal gene transfer of *apr*AB genes [71].

OTU-3 also belongs to the order *Burkholderiales*, with 87% identity to a new genus of the family Thiobacillaceae according to GTDB R95 (closest sequence: *Thiobacillus* sp. UBA2186) (Table 2), as well as with sequences from the family *Hydrogenophilaceae*. Members of the Thiobacillaceae and *Hydrogenophilaceae* families are abundant in hot spring microbial mats [20,29], and their *apr*A sequences [75] have been associated with aerobic and facultative chemolithotrophs isolated from hot springs [79,80].

The remaining *apr*A OTUs were associated with SRBs and were classified outside the phylum *Pseudomonadota*. OTU-4 was most abundant at 61.2 and 63.9 °C, with 97% identity to *Thermodesulfovibrio yellowstonii* (phylum *Nitrospirota*) (Figure 3), which has been commonly found in hot springs [20,75], along with its *apr*A gene sequences [81]. OTU-5, which was highly abundant at 58 and 69.7 °C, was related (97.9% identity) to the abundant hot spring genus *Thermodesulfobacterium thermophilum* DSM 1276 of the phylum *Desulfobacterota* (Figure 3, Table 2) [21,22,24,35,82]. OTU-6, a low abundance SRB in Porcelana, was affiliated (93% identity) with *Desulfofundulus australicus* DSM 11792 (formerly the genus *Desulfotomaculum*, GTDB phylum Bacillota_B; Table 2). This species oxidizes organic compounds along with performing sulfate reduction, while some strains can also reduce sulfite and thiosulfate [83,84]. These bacteria have also been found to dominate at temperatures between 48 and 60 °C in sediments and hot spring microbial mats based on analyses of the 16S rRNA gene and the *apr*A and *dsr*A functional genes [23,29]. In contrast, the 26 *apr*A sequences recovered from the Porcelana metagenomes were distributed among the GTDB phyla *Pseudomonadota*, *Nitrospirota*, Bacillota_A, Bacillota_F, *Planctomycetota*, *Desulfobacterota*, *Cyanobacteriota*, and SZUA-182 (Appendix A). Only seven metagenomic sequences clustered with OTU-1, one with OTU-3 (order *Burkholderiales*), and one with OTU-4 (genus *Thermodesulfovibrio*) (Figure 3). Notably, the complete CDS associated with the phylum *Cyanobacteriota* (88% identity with *Pseudanabaena* sp. SR411) shows a well-documented representation within the phylum (Figure 3) [85] and supports the high activity of this pathway observed in the metatranscriptomes.

As for the *sox*B gene (Figure 4), the two recovered OTUs belong to the phylum *Pseudomonadota* (*Proteobacteria*). The less abundant OTU-2 (Table 2) was affiliated with the order *Rizhobiales* (*Alphaproteobacteria*) and showed 80% identity to *Filomicrobium insigne* CGMCC 1.6497. Neither the species nor the genus have been widely reported in hot springs, but detected at temperatures above 40 °C [86,87]. OTU-1 was affiliated (100% identity) with the hot spring-predominant species *Hydrogenophilus thermoluteolus* (GCF_003574215.1; Figure 4). Members of the family *Hydrogenophilaceae* (order *Burkholderiales*) were also found in Porcelana by *apr*A gene analysis. The presence of the *sox*B and *apr*A genes has been reported in some members of the phylogenetically close genera *Sulfuriferula* and *Thiobacillus* [33,88]; however, the presence of the *apr*A gene in the genus *Hydrogenophilus* has never been reported before [28,89]. In addition, two MAGs recovered at 48 °C were classified by GTDB R95 as a new genus of the family Burkholderiaceae and order *Burkholderiales*. These MAGs recruited one *apr*A and one *sox*B sequence each (Appendix A). Their *apr*A sequences were associated with OTU-1, whereas the *sox*B sequences grouped with the Methylomirabilota clade and the genus *Cupriavidus* (family Burkholderiaceae) (Figure 3), respectively. This could suggest the existence of a new member of the order *Burkholderiales* along the temperature gradient in Porcelana that could present both functional genes.

Furthermore, compared to the clone libraries, Porcelana metagenomes recovered more diverse sequences for the *sox*B gene. These were associated with four different orders of *Alphaproteobacteria* (Appendix A), revealing a high number of SOB that remain to be studied and isolated from hot springs. These *sox*B sequences from *Pseudomonadota* were associated with *Gammaproteobacteria* (eight sequences forming two different clades in the family Burkholderiaceae; Figure 4) and *Alphaproteobacteria* (13 sequences associated with the order *Acetobacterales*, 2 with Rodobacterales and 2 with *Rizhobiales*; Appendix A). Another 16 sequences grouped with the family Thermaceae of the GTDB phylum Deinococcota, while 8 were closely affiliated with (but not part of the same clade as) sequences of the order Rokubacteriales (phylum Methylomirabilota in the GTDB; phylum Candidatus Rokubacteria in the NCBI database) (Figure 4), for which the potential conversion of various sulfur compounds has been suggested [90].

In summary, our results reveal that several members of the phototrophic community from Porcelana hot spring mats, especially, *Pseudomonadota* (*Proteobacteria*), *Chloroflexota*, and *Cyanobacteriota*, are more involved in the sulfur cycle than previously believed (Figure 5). While sulfur was one of the main discriminating factors between different types of microbial communities in different hot springs in Yellowstone National Park, USA [91], and further studies determined a closed sulfur cycle in the undermat of one of the photoautotrophic microbial communities (60 °C) [17], here, we show an important role of this biogeochemical cycle in non-acidic hot spring microbial communities, as has been seen in thermophilic streamer communities and archaeal systems [91] in which the photoautotrophic metabolism is not the main input of carbon and energy. 

In this case, the sulfate assimilative reduction pathway was the most represented in terms of taxonomic abundance and activity. As each of the taxa related to the sulfur cycle has different activity and occurrence along the thermal gradient at the Porcelana hot spring, the observed changes in sulfate and thiosulfate concentrations along the gradient could potentially be explained by the presence of these sulfur metabolizers and their differential activities, with the main primary producers being closely associated to the sulfur cycle, which are also reported to inhabit at different temperatures along the thermal gradient [36]; however, the role of the primary producers in the sulfur cycle of Porcelana hot spring and others non-acidic thermal systems requires further confirmation.

## 4. Conclusions

In this work, we used a combination of metagenomic, metatranscriptomics, and molecular markers to obtain novel information on the diversity of microorganisms involved in the biogeochemical sulfur cycle in microbial mats along a temperature gradient of the neutral pH Porcelana hot spring. The presence of SOB (including highly abundant *Chloroflexota*) and the diversity of SRB in these microbial mats explain the complete reduction and oxidation of inorganic sulfur species in this habitat. In addition to the phyla *Chloroflexota* and *Cyanobacteriota* (previously determined to be the main carbon and nitrogen fixers in this microbial community), which were the most abundant sulfur-related organisms in the Porcelana hot spring at high and low temperatures, respectively, the phylum *Pseudomonadota* (*Proteobacteria*) showed the widest distribution across the thermal gradient. The richness of the SOB and SRB communities remained constant, although their relative abundances varied at different temperatures along the gradient. These differences allow for a decrease in sulfate levels along the gradient from the high-temperature source to the lower-temperature sites downstream, as well as a slight increase in thiosulfate levels along the same gradient. Analysis of diversity indices and phylogenetic composition of sulfur-metabolizing bacteria benefited from the complementary use of different methods and molecular markers, revealing the *apr*A marker in eight phyla and the *sox*B marker in four phyla, including a group of microorganisms with no related sequences in the databases. Recruitment of sequences in MAGs recovered from Porcelana and unbinned contigs further revealed the potential of uncultured microorganisms for both processes. Interestingly, our study in a sulfur-poor non-acidic hot spring such as Porcelana shows similar results to a previous study indicating that metabolic redundancy is relevant in sulfur-rich environments [92], suggesting that sulfur-related metabolic redundancy is widespread in microbial-dominated environments. Taken together, these results are key to further deciphering the role of cultured and uncultured microorganisms in the sulfur cycle, and to better understanding their relationship with primary producers in the complex microbial metabolic networks of hot spring microbial mats.

## Figures and Tables

**Figure 1 microorganisms-11-01803-f001:**
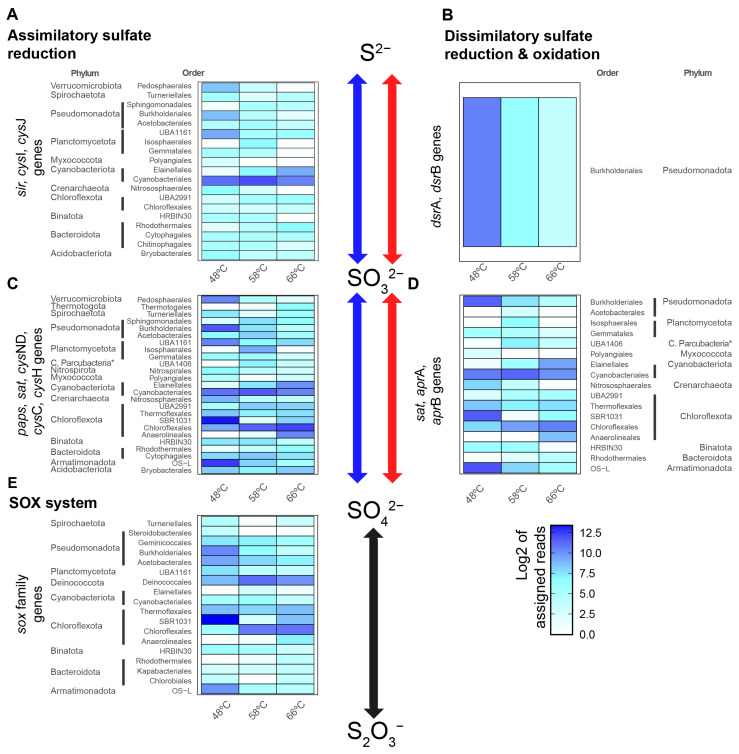
Abundance of sulfur metabolism genes in Porcelana hot spring metagenomes. Genes from the sulfur cycle are grouped based on the pathway ((**A**,**C**) for ASR; (**B**,**D**) for DSR; and (**E**) for Sox) and at which step of the sulfur cycle reaction they are involved ((**A**,**B**) for sulfide to sulfite; (**C**,**D**) for sulfite to sulfate; (**E**) for sulfate to thiosulfate). Orders with their respective phyla are shown on the *Y*-axis and samples are shown on the *X*-axis. The reads counts are normalized by quantiles and Log2-transformed for metagenomes from the three sampled temperatures (48, 58, and 66 °C). Analyzed genes are listed in Appendix A. * C denote “Candidatus”.

**Figure 2 microorganisms-11-01803-f002:**
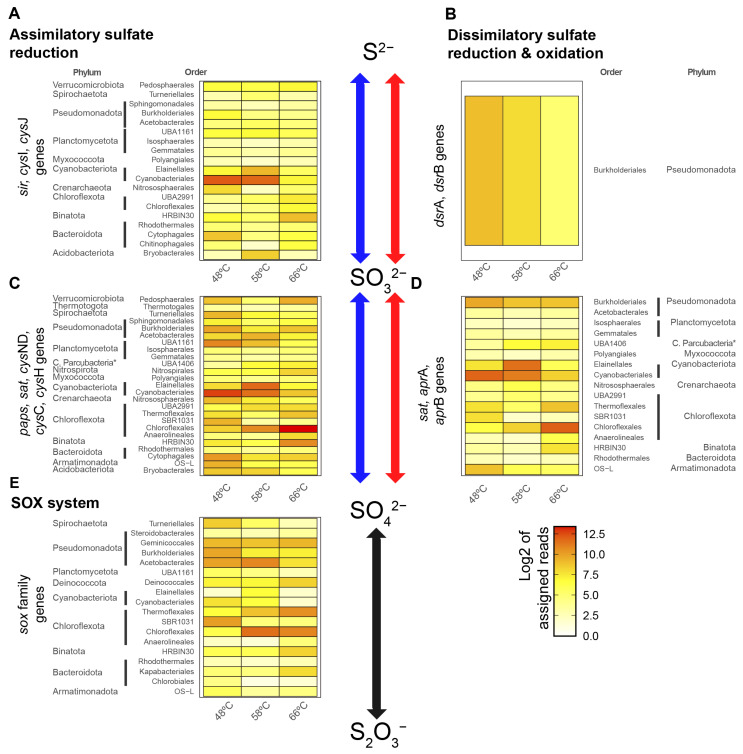
Activity of sulfur metabolism genes in Porcelana hot spring metatranscriptomes. The genes from the sulfur cycle are grouped based on the pathway ((**A**,**C**) for ASR; (**B**,**D**) for DSR; and (**E**) for Sox) and at which step of the sulfur cycle reaction they are involved ((**A**,**B**) for sulfide to sulfite; (**C**,**D**) for sulfite to sulfate; (**E**) for sulfate to thiosulfate). Orders with their respective phyla are shown on the *Y*-axis and samples are shown on the *X*-axis. The read counts are normalized by quantiles and Log2-transformed for metatranscriptomes from the three sampled temperatures (48, 58, and 66 °C). Analyzed genes are listed in Appendix A. * C denote “Candidatus”.

**Figure 3 microorganisms-11-01803-f003:**
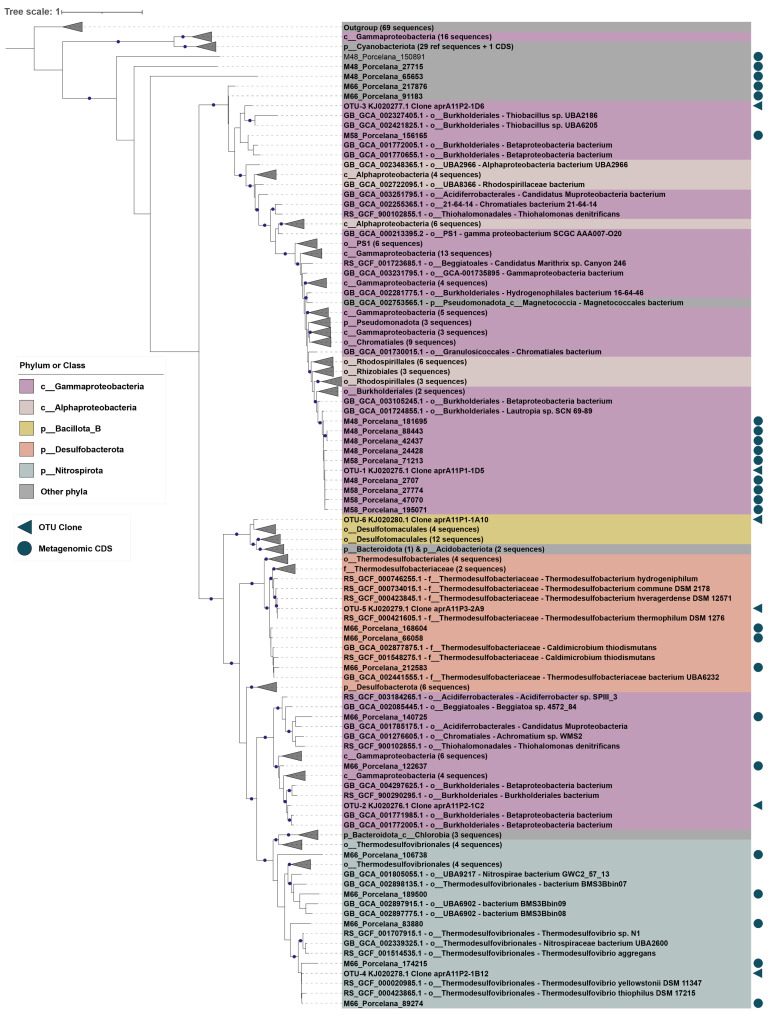
Phylogenetic reconstruction of the *apr*A marker gene. Maximum-likelihood tree reconstruction for the alignment of 307 sequences that include 275 CDS from the GTDB R95 protein database, 26 CDS from the Porcelana metagenomes (blue circles), and 6 clone library *apr*A OTUs (blue triangles). The tree reconstruction was performed with IQtree software using the LG + R9 substitution model and a non-parametric UF-bootstrap support of 1000 replicates. The leaf labels are colored according to GTDB phyla, and specifically by class for sequences from the phylum *Pseudomonadota* (*Proteobacteria*). The tree was rooted in the node between a clade of 69 outgroup sequences (fumarate reductase/succinate dehydrogenase flavoprotein subunit) and a separate clade of 16 *Gammaproteobacteria* sequences and 30 *Cyanobacteriota* sequences. Purple dots in the tree represent > 95% bootstrap support for that node.

**Figure 4 microorganisms-11-01803-f004:**
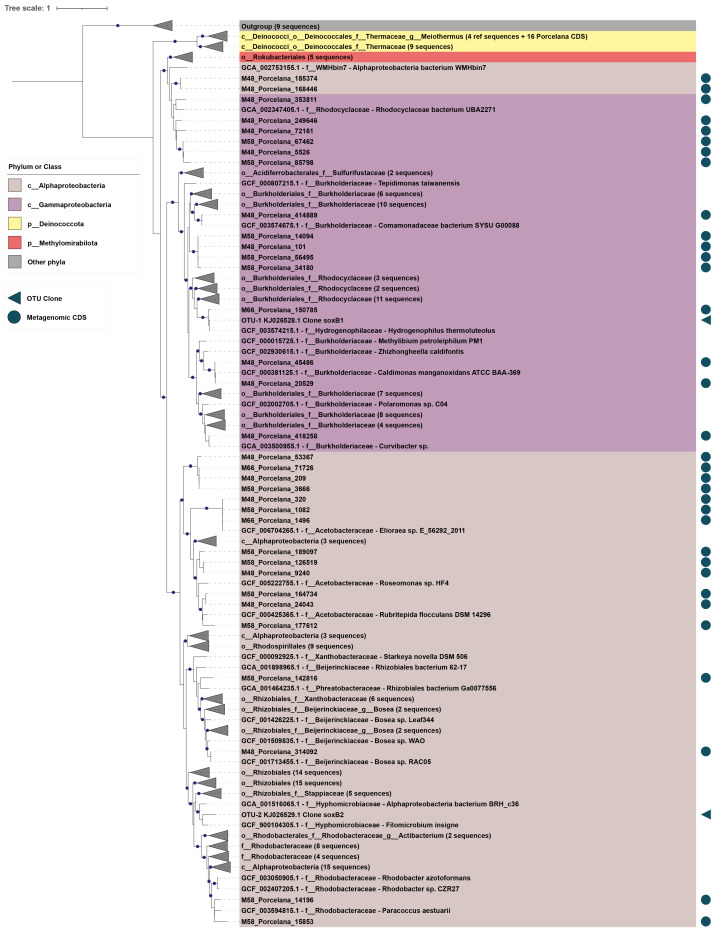
Phylogenetic reconstruction of the *sox*B marker gene. Maximum-likelihood tree reconstruction for the alignment of 246 sequences that include 189 CDS from the GTDB R95 protein database, 50 CDS from the Porcelana metagenomes (blue circles), and two *sox*B OTUs from the clone libraries (blue triangles). The tree reconstruction was performed with IQtree software using the LG + F + R7 substitution model and a non-parametric UF-bootstrap support of 1000 replicates. The leaf labels are colored according to GTDB phyla, and specifically by class for sequences from the phylum *Pseudomonadota* (*Proteobacteria*). The tree was rooted in the node between a clade of 9 outgroup sequences (trifunctional nucleotide phosphoesterase protein YfkN) and the other 246 sequences. Purple dots represent > 95% bootstrap support for that node.

**Figure 5 microorganisms-11-01803-f005:**
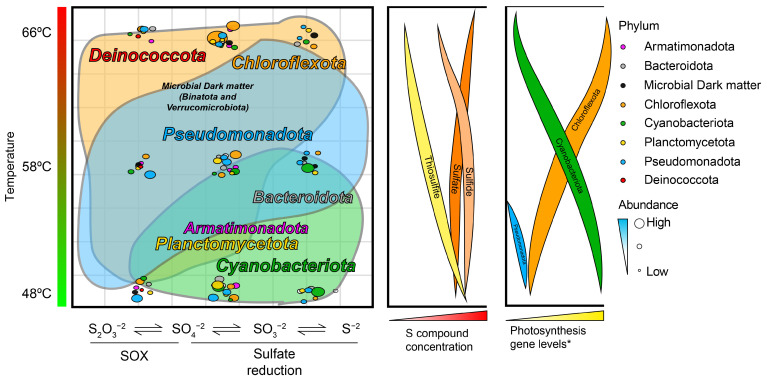
Graphical summary of the microbial community associated with the sulfur cycle at the Porcelana hot spring. In the left panel, a dot plot was built using the gene counts of each sulfur pathway (Appendix A) for 8 different phyla and sulfur cycle steps. The size of the colored circle denotes abundance, and the color denotes the respective phylum. Temperature is shown on the *Y*-axis and sulfur cycle steps on the *X*-axis. The phylum names are written on the *Y*-axis, represented as a function of temperature and the sulfur cycle step where they appeared most relevant. The colored area denoting the most relevant Porcelana sulfur-associated bacteria (*Pseudomonadota*, *Chloroflexota*, and *Cyanobacteriota*) was hand-drawn behind the dots. In the central panel, concentrations of three sulfur species are shown based on field-measured concentrations (Appendix A). In the right panel, the abundance level of photosynthesis-associated genes the of phyla *Cyanobacteria*, *Chloroflexota*, and *Pseudomonadota*. Temperature is shown on the *Y*-axis and concentration on the *X*-axis. * Based on data obtained from Alcamán-Arias et al. [36].

**Table 1 microorganisms-11-01803-t001:** *apr*A and *sox*B OTU richness and diversity parameters in the Porcelana hot spring.

Gene	Diversity Index	Sample 48_2013	Sample 56.8_2011	Sample 58_2013	Sample 61.2_2011	Sample 63.9_2011	Sample 66_2013	Sample 69.7_2011
*apr*A	Simpson (1-D)	0.112	0.056	0.718	0.409	0.383	0.675	0.413
*apr*A	Shannon (H)	0.291	0.149	1.391	0.855	0.802	1.213	0.762
*apr*A	Evenness (J)	0.181	0.135	0.863	0.531	0.498	0.753	0.549
*apr*A	Richness	5	3	5	5	5	5	4
*sox*B	Simpson (1-D)	0.499	0.368	0.32	0.147	0.336	0.313	0.436
*sox*B	Shannon (H)	0.692	0.555	0.504	0.278	0.519	0.492	0.628
*sox*B	Evenness (J)	0.992	0.801	0.721	0.402	0.749	0.71	0.906
*sox*B	Richness	2	2	2	2	2	2	2

**Table 2 microorganisms-11-01803-t002:** Relative abundance of the Porcelana OTUs for the *apr*A and *sox*B genes, and their affiliation with GTDB sequences.

Gene	OTU Clone	Relative Abundance							NCBI Nucleotide ID	GTDB R95 Taxonomy Best Hit ^1^	% Aminoacid Identity	Matched Genome
-	-	Sample 48_2013	Sample 56.8_2011	Sample 58_2013	Sample 61.2_2011	Sample 63.9_2011	Sample 66_2013	Sample 69.7_2011				
*apr*A	OTU-1	0.941	0.971	0.406	0.756	0.774	0.325	0.739	KJ020275.1	o_*Burkholderiales* f_Burkholderiaceae g_SCN-69-89	98.5	GCA_001724855.1
*apr*A	OTU-2	0.008	0.000	0.146	0.000	0.043	0.400	0.000	KJ020276.1	o_*Burkholderiales* f__SG8-39 g__SCGC-AG-212-J23	94.7	GCA_004297625.1
*apr*A	OTU-3	0.008	0.000	0.271	0.022	0.097	0.238	0.000	KJ020277.1	o_*Burkholderiales* f_*Hydrogenophilaceae* g_UBA6918	87.3	GCA_002327405.1
*apr*A	OTU-4	0.034	0.000	0.031	0.089	0.075	0.000	0.043	KJ020278.1	o_Thermodesulfovibrionales f_Thermodesulfovibrionaceae g_Thermodesulfovibrio	97.9	GCF_000020985.1
*apr*A	OTU-5	0.008	0.019	0.146	0.100	0.011	0.025	0.196	KJ020279.1	f_Thermodesulfobacteriaceae g_Thermodesulfobacterium	97.8	GCF_000421605.1
*apr*A	OTU-6	0.000	0.010	0.000	0.033	0.000	0.013	0.022	KJ020280.1	o_Desulfotomaculales f_Desulfovirgulaceae g_Desulfotomaculum_A	93.2	GCF_900129285.1
*sox*B	OTU-1	0.511	0.764	0.800	0.914	0.784	0.811	0.678	KJ026528.1	o_*Burkholderiales* f_Rhodocyclaceae g_Tepidiphilus	100.0	GCF_001418245.1 *
*sox*B	OTU-2	0.489	0.236	0.200	0.086	0.216	0.189	0.322	KJ026529.1	o_Rhizobiales f_Hyphomicrobiaceae g_Filomicrobium	80.4	GCF_900104305.1

^1^ Showing taxonomy from order to genus. Complete taxonomy can be found in Appendix A. * The GTDB taxonomy of the GCF_001418245.1 genome has changed at family level between database versions being classified in f_Hydrogenophilaceae in the R95 and in f_Rhodocyclaceae in older and newer database versions.

## Data Availability

The clone sequences obtained in this study have been deposited in GenBank under accession numbers KJ020275 to KJ020280 for the *apr*A gene; and KJ026528 and KJ026529 for the *sox*B gene. Metagenomes and metatranscriptomes are available under NCBI BioProject accession number PRJNA382437.

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
