# Peer review of "Distribution and Activity of Sulfur-Metabolizing Bacteria along the Temperature Gradient in Phototrophic Mats of the Chilean Hot Spring Porcelana"

_microorganisms, 2023, doi:10.3390/microorganisms11071803_

Round 1

Reviewer 1 Report

The authors have addressed most of my concerns, making the manuscript nicer than before. However, there are still some issues (See below for details) that have to be corrected or revised before the manuscript can be accepted for publication.

 Line 218-236: Much of the content does not highlight the main result and thus should be moved to in the method section and

 line 228 -230: Unclear how to calculate abundance of sulfur metabolism in different microorganisms, without any explanation.

 line 237-302: More rigorously, the evolutionary relationships of DSR proteins can be analyzed to distinguish between oxidative and reductive types. In this way, the sulfur metabolism capacity of different microbial species can be identified. There seems to be no main conclusion to the current results.

Author Response

Dear  Microorganisms Journal Editors and Reviewers,

We appreciate all reviewers’ comments which have contributed to improve the quality of our manuscript. In this new version, we have taken into account the reviewer´s suggestions to discuss the main sulfur-metabolizing organisms at both phylum and order level. In addition, we performed some of the requested analyses, which on the other hand we had already performed before but not included in the manuscript due to the fact that robust results were not obtained because of the reduced number of existing samples.

Below we respond to all reviewers' comments in detail:

Comments and Suggestions for Authors

The authors have addressed most of my concerns, making the manuscript nicer than before. However, there are still some issues (See below for details) that have to be corrected or revised before the manuscript can be accepted for publication.

R: Authors appreciate the comments given by reviewers, as they helped to improve the manuscript.

 Line 218-236: Much of the content does not highlight the main result and thus should be moved to in the method section and

R: We agree with the reviewer and the paragraph was reduced accordingly, moving certain parts to the respective methods section. The paragraph is in lines 167-183 in the current version.

 line 228 -230: Unclear how to calculate abundance of sulfur metabolism in different microorganisms, without any explanation.

R: Thanks for this comment. The explanation was improved in lines 235-238, as well as in the respective methods section in lines 167-183.

 line 237-302: More rigorously, the evolutionary relationships of DSR proteins can be analyzed to distinguish between oxidative and reductive types. In this way, the sulfur metabolism capacity of different microbial species can be identified. There seems to be no main conclusion to the current results.

R: Thanks for this comment. As we aimed for a more general description of the sulfur cycle in metagenomes and metatranscriptomes, we think that the specific information about each of the proteins found in metagenomes was beyond the scope of our manuscript. However, we still think it is valuable information, and we added a sentence explaining that we found microbes belonging to Pseudomonadota (Proteobacteria) that could carry the reverse isoform of the DsrAB protein with their respective reference (lines 284-287). We also added a sentence (lines 486-490) in the conclusion section about the importance of metabolic redundancy, which is not directly related to this comment, but which we found relevant since different members of the bacterial community are part of this biogeochemical cycle.

Reviewer 2 Report

The ms "Distribution and activity of sulfur-metabolizing bacteria along the temperature gradient of Porcelana phototrophic hot spring mats" by Konrad et al is an interesting study of the effect of temperature gradient on the microbial diversity of a hot spring mat. The ms is an intrresting contribution to the sulfur microbial world. Used methods are appropriated for the reached conclusions. My only concern is the lack of consistency in the taxonomic terminology used. Authors mix the names of the new valid phyla with the old nomenclature which I think has to be avoided. 

Comments and suggestion:

- Title, I suggest to introduce the name of the country, Chile, in the title to facilitate its identification.

- Line 28 and along the text. If Chloroflexota is used then Pseudomonadota instead of Poteobacteria should be used too.

- Line 35 and along the text. Eventhough Cyanobacteriota is still a proposal phyla name for Cyanobacteria, taking in consideration that Aharon Oren is the author of the proposal I suggest to use the proposed new nomenclature to have a taxonomically updated ms. 

- Line 61. I think "energy processes" is better than "energetic processes".

- Line 82 and along the text. Use Chlorobiota instead of Cholorobi.

- Figure 1, replace Proteobacteria for Psudomonadora and Cyanobacteria for Cyanobacteriota.

- Fig 2, same as in Fig 1.

- Line 407, replace Firmicutes for Bacillota.

- Figure 5, replace Proteobacteria for Pseudomonadota and Cyanobacteria for Cyanobacteriota. 

Author Response

Dear  Microorganisms Journal Editors and Reviewers,

We appreciate all reviewers’ comments which have contributed to improve the quality of our manuscript. In this new version, we have taken into account the reviewer´s suggestions to discuss the main sulfur-metabolizing organisms at both phylum and order level. In addition, we performed some of the requested analyses, which on the other hand we had already performed before but not included in the manuscript due to the fact that robust results were not obtained because of the reduced number of existing samples.

Below we respond to all reviewers' comments in detail:

Comments and Suggestions for Authors

The ms "Distribution and activity of sulfur-metabolizing bacteria along the temperature gradient of Porcelana phototrophic hot spring mats" by Konrad et al is an interesting study of the effect of temperature gradient on the microbial diversity of a hot spring mat. The ms is an intrresting contribution to the sulfur microbial world. Used methods are appropriated for the reached conclusions. My only concern is the lack of consistency in the taxonomic terminology used. Authors mix the names of the new valid phyla with the old nomenclature which I think has to be avoided.

R: Authors appreciate the positive comments from the reviewer. Taxonomic changes are very recent, and databases changes have been made during the preparation of this manuscript, therefore we have followed the reviewer’s recommendations according to the latest GTDB and NCBI taxonomies, leaving common or classical taxonomies in parentheses when appropriate. We have added an explanatory sentence on lines 157-159. 

Comments and suggestion:

- Title, I suggest to introduce the name of the country, Chile, in the title to facilitate its identification.

R: We appreciate this comment, and the title was changed accordingly.

- Line 28 and along the text. If Chloroflexota is used then Pseudomonadota instead of Poteobacteria should be used too.

R: Thanks for this comment. The new phylum name for Proteobacteria (doi:10.1099/ijsem.0.005056) was changed accordingly in the text, sometimes emphasizing in parenthesis the old/common name.

- Line 35 and along the text. Even though Cyanobacteriota is still a proposal phyla name for Cyanobacteria, taking in consideration that Aharon Oren is the author of the proposal I suggest to use the proposed new nomenclature to have a taxonomically updated ms.

R: Thanks for this comment. Although it seemed that there was an agreement in not changing the Cyanobacteria phylum name, we noted that along with the proposal of Dr. Oren (doi:10.1099/ijsem.0.005528), the GTDB also changed the phylum name in its last version and we changed it accordingly as well.

- Line 61. I think "energy processes" is better than "energetic processes".

R: Fixed. Thanks for this comment.

- Line 82 and along the text. Use Chlorobiota instead of Cholorobi.

R: Fixed. Thanks for this comment.

- Figure 1, replace Proteobacteria for Psudomonadora and Cyanobacteria for Cyanobacteriota.

- Fig 2, same as in Fig 1.

R: Fixed. Thanks for this comment.

- Line 407, replace Firmicutes for Bacillota.

R: Fixed. In the GTDB this phylum is renamed as Bacillota_B.

- Figure 5, replace Proteobacteria for Pseudomonadota and Cyanobacteria for Cyanobacteriota.

R: Fixed. Thanks for this comment.

Reviewer 3 Report

This manuscript includes interesting new findings about the distribution of two sulfur-metabolizing genes, aprA and soxB, in various biomats in hot springs along the temperature gradient focusing on the taxonomic groups of bacteria. The information on the temperature-dependent changes of microbes involved in sulfur metabolism is important for understanding the diversity of the sulfur cycle on Earth and the evolution of sulfur metabolism. However, the description of the manuscript is mainly about the phylum-level analysis. The many phyla are diverse regarding energy metabolism, and dissimilatory sulfur metabolism is not major in most phyla. The reviewer strongly recommends the authors reorganize and shorten the manuscript only focusing on Table 2 and Section 3.4 of “Results and discussion.” The description should more focus on the species, genus, and postulated novel genus levels. The values of “% amino acid identity” in Table 2 imply the description on those levels is possible and appropriate. 

Author Response

Dear  Microorganisms Journal Editors and Reviewers,

We appreciate all reviewers’ comments which have contributed to improve the quality of our manuscript. In this new version, we have taken into account the reviewer´s suggestions to discuss the main sulfur-metabolizing organisms at both phylum and order level. In addition, we performed some of the requested analyses, which on the other hand we had already performed before but not included in the manuscript due to the fact that robust results were not obtained because of the reduced number of existing samples.

Below we respond to all reviewers' comments in detail:

Comments and Suggestions for Authors

This manuscript includes interesting new findings about the distribution of two sulfur-metabolizing genes, aprA and soxB, in various biomats in hot springs along the temperature gradient focusing on the taxonomic groups of bacteria. The information on the temperature-dependent changes of microbes involved in sulfur metabolism is important for understanding the diversity of the sulfur cycle on Earth and the evolution of sulfur metabolism. However, the description of the manuscript is mainly about the phylum-level analysis. The many phyla are diverse regarding energy metabolism, and dissimilatory sulfur metabolism is not major in most phyla. The reviewer strongly recommends the authors reorganize and shorten the manuscript only focusing on Table 2 and Section 3.4 of “Results and discussion.” The description should more focus on the species, genus, and postulated novel genus levels. The values of “% amino acid identity” in Table 2 imply the description on those levels is possible and appropriate.

R: Authors appreciate the reviewer’s comments. In this new version we have further discussed the presence and activities of sulfur-metabolizing microorganisms at the order level in the metagenome/metatranscriptome analysis in section 3.2, including Figures 1 and 2. The phylogenetic analysis concerning aprA and soxB OTUs and CDS are well resolved even at the genus level, and when possible the genera are mentioned.

Although interesting, the values of amino-acid identities with respect to known sequences in the databases do not allow us to propose the novelty of possible sulfur metabolizing microorganisms (e.g., 80 % identity of OTU2 of soxB with respect to a sequence of Filomicrobium). However, when the sequences were binned in MAGs, with genome-wide novelty determined by the GTDB-tk software, these results were discussed further (e. g. lines 425 - 432).

Reviewer 4 Report

This manuscript described on the temperature dependent distribution of microorganism of different sulfur metabolic activity in Porcelana phototrophic hot spring mat. 

In this study, the authors investigated temperature-dependent sulfur-metabolisms in terms of not only metagenomes but also metatranscriptomes. If possible, please insert concentration range of sulfurous.

1)     Based on the background studies, please describe more specifically the issues you want to clarify in this research. In addition, please describe the influence of the obtained results to this field.

2)     Please discuss the advanced points of considering with the background studies of ecological studies on sulfur metabolizing bacteria.

3)     Please estimate bacterial community network on each temperature.

4)     Please estimate relationship environmental parameter and the microbial community by canonical correspondence analysis (CCA) or redundancy analysis (RDA).

Round 2

Reviewer 3 Report

I do not recognize that the authors sufficiently shortened the manuscript as I have strongly recommended in the previous comments. However, since I think some new findings in this manuscript are worth publication in Microorganisms, I agree to be published if another reviewer(s) and the editor think it is appropriate to be published in the present form.

I want to ask the authors to concern the following points.

L61-62: DSR does not use sulfur as a source of electrons. This sentence should be rewritten.

L238-239: The reviewer did not understand the meaning of the sentence. How the assimilatory sulfate REDUCTION pathway can be active of sulfide-to-sulfate OXIDATION pathway.

L265 and many others: “Assimilatory reaction”, rather than ASR, and “dissimilatory reaction”, rather than DSR, are appropriate words since the authors deal with both reduction and oxidation reactions but not only “sulfate reduction” if the reviewer understands the meaning correctly.

Author Response

Dear  Microorganisms Journal Editors and reviewers,

We appreciate the new reviewers’ comments which helped to keep improving the quality of our manuscript. In this new version, we took into account the suggestions of the reviewers to improve the definitions in the direction of reactions, shortening a bit the manuscript and to correct and improve the figures.

Below is our response to all comments:

Reviewer #3

I do not recognize that the authors sufficiently shortened the manuscript as I have strongly recommended in the previous comments. However, since I think some new findings in this manuscript are worth publication in Microorganisms, I agree to be published if another reviewer(s) and the editor think it is appropriate to be published in the present form.

R: The authors again appreciate the reviewer's time and comments. However, the suggestion to change the focus of the manuscript by shortening some specific sections was not fully addressed. In agreement with the editor, we decided after further effort to shorten the manuscript by avoiding redundancies in the Results section (mostly sections 3.1 and 3.2) as much as possible.

I want to ask the authors to concern the following points.

L61-62: DSR does not use sulfur as a source of electrons. This sentence should be rewritten.

R: The authors are grateful for the reviewer's comments. The sentence was rewritten as: "whereas DSR and SOX are mainly associated with energy processes that use sulfur in electron transfer associated processes" in lines 61-62.

L238-239: The reviewer did not understand the meaning of the sentence. How the assimilatory sulfate REDUCTION pathway can be active of sulfide-to-sulfate OXIDATION pathway.

R: The authors are grateful for the reviewer's comments. We apologize for overlooking it, and it was entirely our fault. We have now revised this throughout the manuscript. Although some of the reactions can occur in both directions, we prefer to refer to them as reduction pathways, as most of the literature refers to them in this way. Whereas, with oxidation we only refer to general and not specific processes. This was corrected on lines 232-233 and across the manuscript.

L265 and many others: “Assimilatory reaction”, rather than ASR, and “dissimilatory reaction”, rather than DSR, are appropriate words since the authors deal with both reduction and oxidation reactions but not only “sulfate reduction” if the reviewer understands the meaning correctly.

R: As in the previous comment, we have revised this throughout the manuscript and not only in the line mentioned.

In addition, we mention that some of the reactions associated with sulfur can be of both forms, but that we refer to the reductive form pathways, since they are usually mentioned that way in lines 232-233.

Reviewer 4 Report

The manuscript described on the temperature dependent distribution of microorganism of different sulfur metabolic activity in Porcelana phototrophic hot spring mat has been revised.

If the authors had a little more samples and more data on the necessary parameters to be measured, you could expect to have an excellent analyzed data. It is regrettable.

Major points

・I get the impression that the definition of abundance in Figure 5 is ambiguous. For example, circle size of Pseudomonadata in the SOX reaction is large at any temperature, but Fig.1 and Fig.2 do not give me such impressions.

・If possible, please discuss the impact of the obtained knowledge of this study concerning of non-acidic thermal ecosystem to the background studies on sulfur-metabolizing bacterial communities.

・If possible, please add information of photosynthetic activity in Fig. 5.

Minor point:

• L105: “in situ” should be italics.

Author Response

Dear  Microorganisms Journal Editors and Reviewers,

We appreciate the new reviewers’ comments which helped to keep improving the quality of our manuscript. In this new version, we took into account the suggestions of the reviewers to improve the definitions in the direction of reactions, shortening a bit the manuscript and to correct and improve the figures.

Below is our response to all comments:

The manuscript described on the temperature dependent distribution of microorganism of different sulfur metabolic activity in Porcelana phototrophic hot spring mat has been revised.

If the authors had a little more samples and more data on the necessary parameters to be measured, you could expect to have an excellent analyzed data. It is regrettable.

R: Authors are thankful for the comments given by the reviewer. We agree that more samples would have been beneficial for the study.

Major points

I get the impression that the definition of abundance in Figure 5 is ambiguous. For example, circle size of Pseudomonadata in the SOX reaction is large at any temperature, but Fig.1 and Fig.2 do not give me such impressions.

R: We are REALLY grateful for this particular comment, as it allowed us to find an error in our plotting code for Figs. 1 and 2. We first thought that the counts were diluted in the different orders, but the counts belonging to the different enzymes of the respective pathways were not correctly summed. We regret the mistake which was now rectified, where the orders belonging to Pseudomonadota and other phyla are now correctly shown in Figs. 1 and 2.

If possible, please discuss the impact of the obtained knowledge of this study concerning of non-acidic thermal ecosystem to the background studies on sulfur-metabolizing bacterial communities.

R: Although the sulfur cycle has been described as important in microbial communities of acidic hot springs, here we show that this sulfur cycle is also relevant in non-acidic hot springs with low sulfate levels. In the latter we observe a predominance of assimilation reactions carried out by dominant photoautotrophic organisms, but we also reveal the presence and activity in this cycle of microorganisms along a temperature gradient.  This part is now mentioned in more detail in lines 466-474 (the last paragraph of the discussion).

・If possible, please add information of photosynthetic activity in Fig. 5.

R: Photosynthesis in Porcelana hot spring thermal gradient is sustained by its major primary producers, as reported by Alcaman-Arias et al 2018 (doi: 10.3389/fmicb.2018.02353) and cited several times along the text. Considering that the X-axis is only related to sulfur-related reactions, we thought that more information could overload and confuse the information given in the figure. In order to meet the reviewer comment, we added the  sentence “As each of the taxa related to the sulfur cycle has different activity and occurrence along the thermal gradient at Porcelana hot spring, the observed changes in sulfate and thiosulfate concentrations along the gradient could potentially be explained by the presence of these sulfur metabolizers and their differential activities, with the main primary producers being closely associated to the sulfur cycle, which are also reported to inhabit at different temperatures along the thermal gradient (Alcaman-Arias et al 2018); however, the role of the primary producers in the sulfur cycle of Porcelana hot spring and other non-acidic thermal systems requires further confirmation” in lines 474-481, where to expect to relate sulfur-cycle related information to photosynthetic activity along the thermal gradient. In addition, we added a new line plot in Figure 5 reflecting the information reported by Alcaman-Arias et al 2018, indicating that this graph is based on previously published information.

Minor point:

L105: “in situ” should be italics.

R: Thanks for this comment, it was fixed in line 105.